# Non-Preferred Work and the Incidence of Spinal Pain and Psychological Distress—A Prospective Cohort Study

**DOI:** 10.3390/ijerph181910051

**Published:** 2021-09-24

**Authors:** Eva Skillgate, My Isacson Hjortzberg, Petra Strömwall, Johan Hallqvist, Clara Onell, Lena W. Holm, Tony Bohman

**Affiliations:** 1Musculoskeletal & Sports Injury Epidemiology Center, Department of Health Promotion Sciences, Sophiahemmet University, 114 86 Stockholm, Sweden; my.isacson.hjortzberg@hotmail.com (M.I.H.); petra-stromwall@hotmail.com (P.S.); clara.onell@shh.se (C.O.); 2Unit of Intervention and Implementation for Worker Health, Institute of Environmental Medicine, Karolinska Institutet, 171 77 Stockholm, Sweden; lena.holm@ki.se (L.W.H.); tony.bohman@ki.se (T.B.); 3Department of Public Health and Caring Sciences, Family Medicine, Uppsala University, 752 37 Uppsala, Sweden; johan.hallqvist@pubcare.uu.se; 4School of Health and Welfare, Dalarna University, 791 31 Falun, Sweden

**Keywords:** occupational health, psychological distress, spinal pain, sleep

## Abstract

Mental illness and psychological distress are global concerns. This study aimed to investigate the association between having non-preferred work and the incidence of spinal pain, psychological distress, and spinal pain with concurrent psychological distress, and if associations are modified by sleep disturbance. A prospective study of 4285 participants 23–62 years old was conducted, from years 2007 to 2010. Participants reported their work situation as preferred/non-preferred regarding profession/workplace with a high/low possibility to change. Psychological distress was measured with the General Health Questionnaire 12 and spinal pain with questions about neck/back pain. Binominal regression analyses calculated relative risk (RR) with 95% confidence interval (CI). Non-preferred work with a low possibility to change was associated with a higher incidence of spinal pain (RR 1.8; 95% CI 1.2–2.6) and psychological distress (RR 1.8; 95% CI 1.4–2.4) compared to preferred work. The RR was 1.4 (95% CI 0.9–2.1) for spinal pain and 1.3 (95% CI 1.0–1.7) for psychological distress among those with a high possibility to change. Non-preferred work yielded a higher incidence of spinal pain with concurrent psychological distress (RR 1.9; 95% CI 1.0–3.7). Sleep disturbance did not modify associations. A replication based on newer data is needed to confirm the results. In conclusion, non-preferred work is associated with a higher incidence of spinal pain and psychological distress, especially if the possibility to change job is low.

## 1. Introduction

Mental illness and musculoskeletal conditions, including spinal pain, are issues of global public health concern and the leading causes for years lived with disability in high- and middle-income countries [1]. In 2016, 543 million suffered from depression and anxiety [2] and, in 2019, an estimated 790 million people suffered from non-specific neck and low back pain globally [3]. The prevalence of spinal pain is expected to further increase due to a global increase in life expectancy [4]. Furthermore, mental illness and spinal pain are the two primary reasons for sick leave in Sweden [5], as well as in other European countries [6].

Psychological distress is characterized by a continuum of mild to moderate depressive and anxiety symptoms with insufficient severity to attract a formal diagnosis but may develop into mental illness of greater severity if unrecognized [7,8,9]. Spinal pain and psychological distress often occur concurrently, and this co-morbidity does not seldomly result in a less successful recovery and long-term inconvenience, in comparison to having only one of the conditions [10,11]. Both conditions are associated with an individual, financial, and socioeconomic burden and studying these conditions collectively is, hence, of importance [12].

Several risk factors are involved in the development of spinal pain and psychological distress. Old age, a low level of education, poor psychosocial environment and obesity are characteristics associated with chronic back pain [13]. Furthermore, the working environment may play a substantial role in the development of spinal pain [14,15] and an inverse correlation between job satisfaction and back pain intensity has been found [16]. A systematic review and meta-analysis identifying individual worker traits among office workers and physical environmental factors found that a low satisfaction with the workplace environment could be a risk factor for developing non-specific neck pain [17].

Moreover, experiencing high demands, a low possibility to influence the work situation and low management support can contribute to the development of depressive symptoms [18]. Additionally, previous studies have found that being in a non-preferred workplace and occupation (i.e., being “locked-in”) is associated with perceiving high levels of stress, psychological distress, low job satisfaction [19], long-term sick-leave [20] and impaired well-being [21]. In a recent study on Swedish workers [22], poor self-rated general and mental health increased with an increased degree of precariousness at work, indicating that working situations have an impact on workers’ health. Similarly, a study including data from 35 European countries found high precariousness at work to associate with a bad self-reported health status, anxiety, fatigue, and musculoskeletal pain [23].

Sleep disturbance has been identified as playing an important role for the development of spinal pain, likely through the impairment of physiological processes that may contribute to maintaining chronic pain [14,15,24,25,26]. Additionally, the association between sleep disturbance and psychological distress is established [27]. Interpersonal support, equal treatment and the possibility to control the working situation was associated with less sleep disturbance in a systematic review [28]. Conversely, high work demands and a mentally strenuous job were associated with a higher prevalence of sleep disturbances. Given that sleep disturbance is associated with the working environment, spinal pain and psychological distress, it is reasonable to assume that sleep disturbance may modify associations between these conditions.

Several studies have found an association between the working environment, spinal pain and psychological distress. However, to the best of our knowledge, no published studies include aspects of both spinal pain and psychological distress, concurrently, in relation to the working situation. Since work-related mental and physical demands may be of importance for the development of spinal pain and psychological distress, the concept of having ‘non-preferred work’ is developed in this study. A non-preferred work is, here, defined as reporting working in a non-preferred profession and/or non-preferred workplace. The study aims to investigate the association between having a non-preferred work and the incidence of spinal pain, psychological distress and spinal pain with concurrent psychological distress, and to examine if potential associations are modified by sleep disturbance.

## 2. Materials and Methods

### 2.1. Study Design and Sample

This prospective cohort study was based on the Stockholm Public Health Cohort (SPHC). The SPHC is set within the framework of the Stockholm County Council public health surveys [29] in which questionnaires were sent to the general population between 18 and 84 years old and randomly selected after stratification for gender and residential area. The subsample of the SPHC used in this study constituted of information from a first questionnaire in 2002, sent to 50,067 individuals, with follow-ups in 2007 and 2010. The questionnaires surveyed participant characteristics as well as lifestyle, health and profession.

Figure 1 summarizes the recruitment process of the study sample. Information from the SPHC survey in 2007 was used as baseline since the main exposure was measured in that questionnaire. Included in this study were individuals who responded to the SPHC questionnaires in 2002 and 2007, aged 23–62 years at baseline. The upper age limit was set because the most common retirement age in Sweden was 65 years and the participants should still be of working age at the year of follow-up. Further, included participants who answered no to the question; “Have you had neck and/or back pain for periods longer than seven days in a row during the past five years?”, did not have psychological distress (a sum score of less than 3 on the General Health Questionnaire 12 (GHQ-12), details below) [30], and had complete information about the exposure at baseline.

### 2.2. Exposure and Potential Effect Modifier

The 2007 survey questions used to define the exposure are presented in Table 1.

Job experience was categorized into three categories according to the following:-Preferred work was equal to answering “Yes” (a) to questions 1 and 2, regardless of the answer to question 3;-Non-preferred work with a high possibility to change was equal to answering “No” (b) to question 1 and/or “No” (b) to question 2 and answering “Very good” (a) or “Good” (b) to question 3;-Non-preferred work with low possibility to change was equal to answering “No” (b) to questions 1 and/or 2 and “Poor” (c) or “Very poor” (d) to question 3.

Sleep disturbance was assessed with the question; “Do you have trouble sleeping?”. Answers were dichotomized into “no sleep disturbance” (“No”) and “sleep disturbance” (“Yes, somewhat”/“Yes, severe”).

### 2.3. Outcomes

The outcomes for spinal pain and psychological distress were measured in the follow-up questionnaire in 2010. Participants answering “Yes” to one or both of the following questions and in addition having reported the pain to restrict their work capacity and daily activities to some or to a high degree were defined as having spinal pain: (a)“During the past 6 months, have you felt pain in your upper back or neck at least a couple of days per week?” (yes/no);(b)“During the past 6 months, have you felt pain in your lower back at least a couple of days per week?” (yes/no).

Psychological distress was assessed with the General Health Questionnaire 12 (GHQ-12) [30]. The GHQ-12 has previously been validated for use in the Swedish working population with a multivariate analysis using merged data from three studies, showing to be a useful screening instrument measuring mental health, and is stable between men and women as well as between unemployed and employed individuals [31,32]. The GHQ-12 is a commonly used self-reporting instrument to measure psychological distress with high validity and is reliable, also, when used with relatively long intervals between assessments [33,34]. The level of psychological distress in this study was measured using the standard 0-0-1-1 bi-modal scoring of GHQ-12 on the four answering alternatives “Better than usual”, “Same as usual”, “Worse than usual” and “Much worse than usual”, where a sum score of less than 3 indicated no psychological distress and a score of 3 or more indicated psychological distress [34]. Participants who fulfilled both the criteria for spinal pain and psychological distress were defined as having spinal pain with concurrent psychological distress.

### 2.4. Potential Confounders

Potential confounders were chosen in accordance with previous research, clinical considerations and availability, after a careful discussion about if they could possibly constitute intermediators or colliders. The potential confounders were sex, age, socioeconomic status, frequency of alcohol consumption in past 12 months, exercise habits in past 12 months, long-term illness, working hours and household work in past 12 months.

### 2.5. Statistical Analysis

Associations between reporting non-preferred work and the incidence of spinal pain, psychological distress or spinal pain with concurrent psychological distress were calculated using general linear models with a binomial distribution and a log-link. Results were presented as crude and adjusted relative risk (RR) with 95% confidence interval (CI). Analyses were stratified by sleep disturbance in order to assess if the association between the exposure and outcomes was modified by sleep. Due to that only few participants developed spinal pain with concurrent psychological distress, the two exposure categories “non-preferred work, with high possibility to change” and “non-preferred work, with low possibility to change” were merged into one category, denoting “non-preferred work” in the stratified analyses. The questions defining spinal pain in 2007 referred to pain for the previous five years. Therefore, a sensitivity analysis was performed using a sample based only on the answers about spinal pain one year back in time, to assess if information bias may have influenced the results. Post-hoc power analyses were performed with the G*Power software using generated sample sizes and crude estimates for the different outcomes with a one-tailed alpha level of 0.05.

## 3. Results

Baseline characteristics of the analyzed sample are presented in Table 2. The mean age was 41 years and 54% of the participants were women. Eighty-one percent reported having preferred work, 11% reported having non-preferred work with a high possibility to change and 8% reported having non-preferred work with a low possibility to change. Additionally, 28% reported sleep disturbance.

Due to limited statistical power in some analyses, only one confounder, age, was considered a confounder in the associations between non-preferred work and spinal pain and psychological distress, respectively, and the analyses were adjusted accordingly. Table 3 shows crude and adjusted RR and 95% CI for developing spinal pain and psychological distress among participants reporting a non-preferred work in comparison to those reporting a preferred work at baseline. Due to low statistical power in the analyses of the outcome spinal pain with concurrent psychological distress, only crude values were presented. When comparing participants reporting non-preferred work with a low possibility to change with those reporting preferred work, the RR was 1.8 (95% CI 1.2–2.6) for spinal pain and 1.8 (95% CI 1.4–2.4) for psychological distress. When comparing participants reporting non-preferred work with a high possibility to change with those reporting preferred work, the RR was 1.4 (95% CI 0.9–2.1) for spinal pain and 1.3 (95% CI 1.0–1.7) for psychological distress. The RR for spinal pain with concurrent psychological distress among participants reporting non-preferred work, regardless of the possibility to change, was 1.9 (95% CI 1.0–3.7). The sensitivity analysis of the cohort of participants free from spinal pain only one year back in time, showed similar results; RR 1.4 (95% CI 1.0–1.9) and 1.7 (1.2–2.3) for spinal pain and RR 1.5 (95% CI 1.3–1.8) and 2.1 (1.8–2.5) for psychological distress (Not in any table).

Table 4 shows crude and adjusted RR and 95% CI of the analysis stratified for sleep disturbance, indicating that there was no effect measure modification of the associations from sleep disturbance.

The post hoc sample size calculations showed a probability of 0.54 for identifying an RR of 1.3 for the association between non-preferred work with a high possibility to change and spinal pain, and of 0.99 for the association between non-preferred work and a low possibility to change and spinal pain. For the outcome, psychological distress was the corresponding power of 99% for identifying an RR of 1.7, and 100% for identifying an RR of 1.8; for the combined outcome, power was 69% to identify an RR of 1.9 for non-preferred work.

## 4. Discussion

The purpose of this prospective cohort study was to investigate the association between non-preferred work and the incidence of spinal pain, psychological distress and spinal pain with concurrent psychological distress, and to explore if potential associations were modified by sleep disturbance. 

The results showed a higher incidence of spinal pain and psychological distress, respectively, among participants reporting non-preferred work, with a low possibility to change, in comparison to participants reporting preferred work. There were, also, indications of a higher incidence of spinal pain and psychological distress for participants reporting non-preferred work, with a high possibility to change. Similar results were found for participants reporting having non-preferred work, regardless of the possibility to change, and spinal pain with concurrent psychological distress, although this result was not statistically significant. 

The findings align with previous studies examining the association between job situation and developing spinal pain and/or psychological distress [35,36,37,38]. Loghmani et al. reported, in a cross-sectional study, that people who did not like their workplace experienced higher pain intensity compared to those who did like their job [16]. Although Loghmani et al. measured job satisfaction with a more extensive instrument than in the current study, no conclusions about causal associations could be drawn due to the cross-sectional design. Additionally, the results in our study are in line with results published by Muhonen (2010), Stengård et al. (2016) and Fahlen et al. (2009), assessing the phenomenon of feeling “locked-in” at work, where being in a non-preferred occupation and workplace impaired multiple aspects of well-being, including psychological and physiological health outcomes as well as sick-leave. Muhonen defined the locked-in phenomenon as having a non-preferred workplace or occupation, whereas being double locked-in was defined as having both a non-preferred workplace and occupation. The exposure was measured with two questions about the preferred workplace and occupation, respectively [19]. These questions were similar to the ones used to measure the exposure in this study. Notably, Muhonen presented preferred workplace and occupation both separately and together, whereas preferred or non-preferred work in this study included both the workplace and/or occupation without distinguishing between their different contributions to spinal pain and/or psychological distress. Moreover, our results are in line with findings by Canivet et al. (2017), who found that having a non-desired occupation was associated with a higher incidence of poor mental health [39].

The stratified analyses indicated that sleep disturbance did not modify the association between job experience, spinal pain and psychological distress. The results were somewhat in accordance with two previous studies finding that sleep disturbance did not modify the association between job strain and troublesome low back pain [15], but between job strain and neck pain [14].

The prospective design, a large sample size and thorough consideration of potential confounding strengthen the validity of the study. Additionally, the outcomes were measured with valid and commonly used self-reporting instruments. However, some limitations need to be considered. As presented in Table 3 and Table 4, there were only a few cases in the analyses of associations between reporting non-preferred work and spinal pain with concurrent psychological distress. Results should be interpreted with caution. Although few cases question the study’s power, still, the large sample size and study design were unique and generated hypotheses about the role of job exposure condition of significance for global public health. Moreover, although several potential confounding factors were considered, the risk of unmeasured or residual confounding cannot be overlooked.

There was a risk of selection bias due to the nature of the cohort. It was formed in 2002 for this study, but we used the follow-up in 2007 as a baseline, when the exposure was measured. Loss to follow-up was present in this step as well as at follow-up in 2010. Additionally, questions regarding job experience were not previously validated, but developed for use in the SPHC, and a risk for a misclassification of the exposure was present. Sleep disturbance was assessed with one question in the questionnaire from 2007. The question was based on participants’ subjective assessment of what it is to have difficulties sleeping, and there was a risk for a potential non-differential misclassification. It may also have influenced the stratified analysis resulting in no modification. In both cases, the most probable effect would be a dilution of the association between the exposure and outcome.

The fact that the data were collected more than 10 years ago, may be considered a limitation. Since the aim was to investigate associations between exposures and outcomes, and not to report on the occurrence, we do not think this had an impact on our conclusions. We had no reason to believe that the associations would have changed over time. Nevertheless, the study shall be replicated based on newer data to confirm the results.

One of the inclusion criteria was being between 23 and 62 years old. That limit was set due to the most common Swedish retirement age of 65 years. However, some participants may have had an early retirement and not been exposed during the last years of the follow-up. Moreover, the exposure may have changed during follow-up for other reasons as a change of employment and work duties. This might have resulted in a dilution of the associations in our study.

In this study sample, the cohort at risk for spinal pain included participants with spinal pain less than seven days in a row. Although the cohort was considered pain-free at baseline, subjects may have had spinal pain but with less severity than our definition. When defining our healthy cohort, there was a risk of misclassification due to difficulties in remembering spinal pain five years back in time at baseline in 2007. However, when performing the analyses in a cohort of participants free of spinal pain only one year back, the results did not change much. Therefore, the way the cohort at risk for spinal pain was created was not considered to be a limitation, but rather a strength, considering the recurrent nature of spinal pain. Additionally, the definition of spinal pain was different between baseline and follow-up. To the best of our knowledge, this study was the first of its kind to investigate if non-preferred work was associated with an increased incidence of spinal pain, psychological distress and spinal pain with concurrent psychological distress. This study contributes to the mapping of risk factors for spinal pain, and psychological distress which is essential for the possibility of preventing these conditions. The results of this study indicate that the workplace could be important for health outcomes and that there may be a need to facilitate job changes and re-education in order to find a preferred work which possibly could prevent spinal pain and psychological distress.

## 5. Conclusions

Non-preferred work seems to be associated with a higher incidence of developing spinal pain, psychological distress and spinal pain with concurrent psychological distress, especially if the possibility to change job is low. However, it is worth mentioning that larger studies are needed to further confirm and support the results. The effect was similar in those with and without sleep disturbances.

## Figures and Tables

**Figure 1 ijerph-18-10051-f001:**
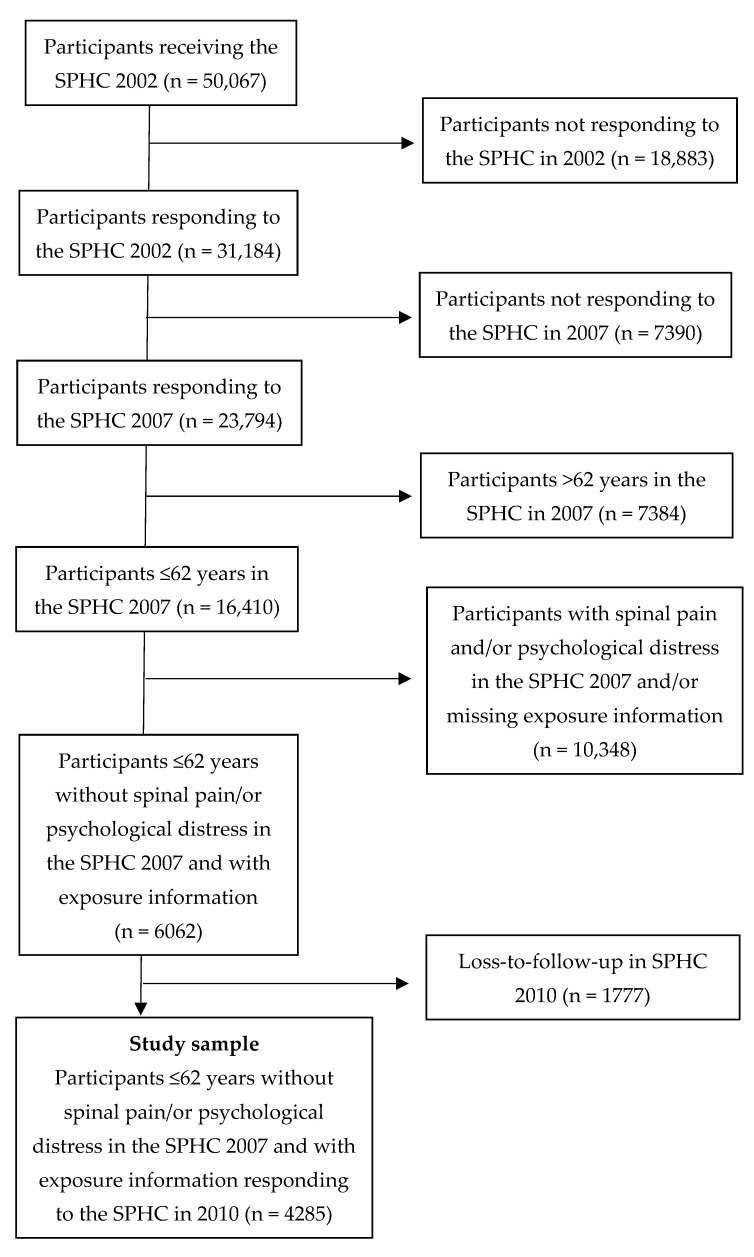
Flowchart of study participant inclusion and follow-up.

**Table 1 ijerph-18-10051-t001:** Questions to survey the exposure job experience.

Questions		Answer Alternatives			
		a	b	c	d
1	Are you currently working in a profession that you believe is right for you?	Yes	No	-	-
2	Are you working at a workplace that is right for you?	Yes	No	-	-
3	How do you view your possibility to change job?	Very good	Good	Poor	Very poor

**Table 2 ijerph-18-10051-t002:** Baseline characteristics of study sample categorized by exposure (n = 4285).

Variable	All, n (%)	Preferred Work, n (%)	Non-Preferred Work, High Possibility to Change, n (%)	Non-Preferred Work, Low Possibility to Change, n (%)
All	4285 (100)	3462 (80.8)	476 (11.1)	347 (8.1)
Women	2298 (53.6)	1829 (79.6)	272 (11.8)	197 (8.6)
Age, mean (SD)	46 (10)	47 (10)	39 (10)	48 (10)
Country of birth				
Sweden	3739 (87.6)	3059 (81.8)	418 (11.2)	262 (7.0)
Elsewhere	527 (12.4)	391 (74.2)	52 (9.9)	84 (15.9)
Socioeconomic status ^a^				
Blue-collar workers	943 (22.7)	737 (78.2)	108 (11.4)	98 (10.4)
White-collar workers	3069 (74.1)	2495 (81.3)	345 (11.2)	230 (7.5)
Self-employed/unclassified	131 (3.2)	113 (86.3)	11 (8.4)	7 (5.3)
Sleep disturbance	1220 (28.5)	950 (77.9)	133 (10.9)	137 (11.2)
Alcohol consumption past 12 months				
Never	205 (4.8)	156 (76.1)	27 (13.2)	22 (10.7)
>1–4 times/month	2441 (57.2)	1946 (79.7)	300 (12.3)	195 (8.0)
≥2 times/week	1623 (38.0)	1346 (82.9)	148 (9.2)	129 (8.0)
Exercise past 12 months				
<1 h/week	1539 (36.2)	1259 (81.8)	149 (9.7)	131 (8.5)
1–2 h/week	1287 (30.3)	1045 (81.2)	141 (11.0)	101 (7.8)
2–3 h/week	755 (17.8)	598 (79.2)	97 (12.8)	60 (7.9)
>3 h/week	671 (15.8)	535 (79.7)	86 (12.8)	50 (7.5)
Long-term illness	747 (17.7)	586 (78.5)	85 (11.4)	76 (10.2)
Working hours				
>45 h/week	746 (18.4)	650 (87.1)	48 (6.4)	48 (6.4)
36–45 h/week	2650 (65.3)	2116 (79.9)	315 (11.9)	219 (8.3)
20–35 h/week	553 (13.6)	437 (79.0)	63 (11.4)	53 (9.6)
1–19 h/week	82 (2.0)	51 (62.2)	23 (28.1)	8 (9.8)
Other working hours	28 (0.7)	22 (78.6)	4 (14.3)	2 (7.1)
Household work past 12 months				
<1 h/day	1283 (30.0)	1024 (79.8)	152 (11.9)	107 (8.3)
1–2 h/day	2065 (48.3)	1676 (81.2)	231 (11.2)	158 (7.7)
>2 h/day	931 (21.7)	756 (81.2)	93 (9.9)	82 (8.8)

^a^ Socioeconomic status, for the economically active population based on current occupation and education, for the inactive population based on previous occupation, current education or the occupation of spouses.

**Table 3 ijerph-18-10051-t003:** Associations between non-preferred work and the incidence of spinal pain, psychological distress and spinal pain with concurrent psychological distress (n = 4285).

Outcomes	Cases/Total	Crude RR (95% CI)	Adjusted RR ^a^ (95% CI)
Spinal pain			
Preferred work	146/3414	1	1
Non-preferred work, high possibility to change	27/472	1.3 (0.9–2.0)	1.4 (0.9–2.1)
Non-preferred work, low possibility to change	26/342	1.8 (1.2–2.7)	1.8 (1.2–2.6)
Psychological distress			
Preferred work	309/3375	1	1
Non-preferred work, high possibility to change	71/460	1.7 (1.3–2.1)	1.3 (1.0–1.7)
Non-preferred work, low possibility to change	54/330	1.8 (1.4–2.3)	1.8 (1.4–2.4)
Spinal pain and psychological distress			
Preferred work	27/3331	1	
Non-preferred work	12/784	1.9 (1.0–3.7)	

^a^ Adjusted for age.

**Table 4 ijerph-18-10051-t004:** Associations between non-preferred work and the incidence of spinal pain, psychological distress and spinal pain with concurrent psychological distress, stratified by sleep disturbance (n = 4285).

Outcomes	No Sleep Disturbance			Sleep Disturbance		
	Cases/ Total	Crude RR (95% CI)	Adjusted RR ^a^ (95% CI)	Cases/Total	Crude RR (95% CI)	Adjusted RR ^a^ (95% CI)
Spinal pain						
Preferred work	80/2473	1	1	65/935	1	1
Non-preferred work	31/547	1.8 (1.2–2.6)	1.8 (1.2–2.7)	22/264	1.2 (0.8–1.9)	1.2 (0.8–2.0)
Psychological distress						
Preferred work	207/2444	1	1	102/925	1	1
Non-preferred work	78/535	1.7 (1.3–2.2)	1.5 (1.1–1.9)	46/252	1.7 (1.2–2.3)	1.5 (1.1–2.0)
Spinal pain and psychological distress						
Preferred work	15/2415	1		12/910	1	
Non-preferred work	6/533	1.8 (0.7–4.6)		6/248	1.8 (0.7–4.8)	

^a^ Adjusted for age.

## Data Availability

The data presented in this study are available on request from the corresponding author. The data are not publicly available due to protection of personal information.

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
