# Peer review of "Non-Preferred Work and the Incidence of Spinal Pain and Psychological Distress—A Prospective Cohort Study"

_ijerph, 2021, doi:10.3390/ijerph181910051_

Round 1
Reviewer 1 Report
Once again congratulations for great work. I do not have anything to add.
Author Response
Thank you.
Reviewer 2 Report
In this manuscript the authors seek to determine if non-preferred work is associated with both the incidence of spinal pain and psychological distress. A strength of the study is that it utilizes a large prospective cohort study. I have a few minor suggestions and am seeking some minor clarifications from the authors:
Tense issues noted throughout manuscript.
Keywords: would add either “spinal pain” or “lower back pain” as a keyword
Line 36: For “spinal pain” are the authors referring to “lower back pain”? I have seen the latter more commonly used but perhaps in Europe if spinal pain is the preferred term, it is fine to leave it alone.
Lines 46-48: This sentence is worded confusingly. I had to re-read it several times to grasp what it was trying to say.
Lines 102-104: Why was the upper limit not set to age 66 then if the retirement age is 65? I see the authors provide a more detailed explanation in the Discussion (lines 330-331) but this would be more appropriate in the Methods section.
Table 1: What is the internal reliability consistency (Cronbach’s alpha) for the items that were used to measure preferred work and non-preferred work (both in high and low possibility to change settings) to show the items for the construct were indeed measuring what was intended?
Table 2: How were blue collar vs. white collar workers defined? Also these terms are being used less often now because some may perceive them to be derogatory.
Lines 324-328: This paragraph looks 1.5 spaced for some reason.
Author Response
- We have done a spell check by an English native. Changes are visible through Track Changes
In this manuscript the authors seek to determine if non-preferred work is associated with both the incidence of spinal pain and psychological distress. A strength of the study is that it utilizes a large prospective cohort study. I have a few minor suggestions and am seeking some minor clarifications from the authors:
Tense issues noted throughout manuscript.
Keywords: would add either “spinal pain” or “lower back pain” as a keyword
- We agree and have changed “Musculoskeletal pain” to “spinal pain”
Line 36: For “spinal pain” are the authors referring to “lower back pain”? I have seen the latter more commonly used but perhaps in Europe if spinal pain is the preferred term, it is fine to leave it alone.
- In Europe we use “low back pain” “upper back pain” and “neck pain” depending on what location that is studied. If all areas are studied – as in this article – we use the word Spinal Pain. The exact definition of spinal pain is described under the heading Methods/Outcome.
- The comment has not led to any changes in the manuscript
Lines 46-48: This sentence is worded confusingly. I had to re-read it several times to grasp what it was trying to say.
- We have tried to clarify in the revised manuscript
Lines 102-104: Why was the upper limit not set to age 66 then if the retirement age is 65? I see the authors provide a more detailed explanation in the Discussion (lines 330-331) but this would be more appropriate in the Methods section.
- We did not want to include persons that were not working, as might be the case when you are 66. We have changed the wording slightly in the Method section. What is written in the discussion about this topic is a discussion about what impact this choice may have had on the results, which we consider shall be kept in the Discussion.
Table 1: What is the internal reliability consistency (Cronbach’s alpha) for the items that were used to measure preferred work and non-preferred work (both in high and low possibility to change settings) to show the items for the construct were indeed measuring what was intended?
- Thank you for this comment. Unfortunatly we cannot see how information about the Cronbach’s alpha would improve our manuscript. If the reviewer wants us to check “if the items for the construct were indeed measuring what was intended” (validity) we argue that this is not a suitable method for this. It provides no evidence for as whether or not the items measure the construct as it claims to measure, but the internal consistency. Internal consistency is not a parameter of validity (de Vet HCW, Terwee CB, Mokkink LB, Knol DL. Measurement in Medicine : A Practical Guide. Cambridge: Cambridge : Cambridge University Press; 2011.Chapter 4.5.3, page 84, second paragraph)". Further – we do not think a measure of consider internal consistency to be relevant in this case. The measures of preferred and non-preferred work are not unidimensional scales, but rather scales that shall not measuring the same thing. The Cronbach’s alpha shall only be used for unidimensional scales (de Vet HCW, Terwee CB, Mokkink LB, Knol DL. Measurement in Medicine : A Practical Guide. Cambridge: Cambridge : Cambridge University Press; 2011.Chapter 9.9.3, page 298, second paragraph"
- The comment has not led to any changes in the manuscript
Table 2: How were blue collar vs. white collar workers defined? Also these terms are being used less often now because some may perceive them to be derogatory.
- Thank you for this comment. We used the Swedish Socioeconomic Classification, developed by Statistics Sweden, which divides individuals in six socioeconomic categories, according to their current occupational status. Blue collar workers were defined as unskilled workers and skilled workers, and White collar workers were non-manual employees, intermediate non manual employees, higher non-manual employees.
- The comment has not led to any changes in the manuscript
Lines 324-328: This paragraph looks 1.5 spaced for some reason.
- Yes we have noticed that but it seems to be something wrong with the template.
- The comment has not led to any changes in the manuscript
This manuscript is a resubmission of an earlier submission. The following is a list of the peer review reports and author responses from that submission.
Round 1
Reviewer 1 Report
The article titled, Non-preferred work and the incidence of spinal pain and psychological distress – a prospective cohort study is interesting and quite original.
It offers data that support the need to go deeper into the knowledge related to workers preferences and well-being. This is a relevant objective that need to be treated more widely in order to contribute to anticipate and reduce the risk of getting sick as a result of inapropiated decisions and habit at the workplace.
Reviewer 2 Report
I would like to congratulate authors as it was really a pleasure to read the paper. It is very interesting, with a good flow of the text. The paper refers to great research material (e.g. sample size) which makes it really valuable. Appropriate methods of analysis were applied, results are well-presented, with credible description of limitations.
There is not much what I can advise to improve the paper.
One technical issue – Part Exposure and potential effect modifier. Can you provide a figure (a kind of tree maybe) to illustrate this part? It would be much easier to follow the text and organize understanding.
One general issue – I can acknowledge the importance of the study for general public although I miss a short paragraph, maybe a short review of the state of the art – can Swedish characteristics regarding for example primary reasons for sick leave be an universal picture – maybe not for the world, but at least for Europe? You wrote: In a recent study on Swedish workers [19], poor self-rated general and mental health increased with increased degree of precariousness at work, indicating that working situation have an impact on workers’ health. Is it similar in other European countries? I can realize that international background is not the scope of the study but if the reasons for starting the topic can have a kind of universal character then your study is important to be followed up in other countries. To sum up, I would recommend adding a short paragraph indicating if some aspects of your study are similar in other (at least European) countries.
Reviewer 3 Report
Thank you for giving me the opportunity to review the article. The author conducted a study focusing on the non-preferred work and the incidence of spinal pain and psychological distress. The topic is socially important, but the there are fundamental problems in the manuscript. Therefore, the reviewer thought that the manuscript cannot be published in the journal IJERPH. I left my major comments for future submission to other journals below.
Comments:
Abstract:
- The authors should mention about the study period in the abstract.
- The authors used the data which obtained over 10 years ago. This is a major limitation of this study. Therefore, the authors should mention about the needs of future investigations using an updated data.
Introduction:
- The authors should cite more updated information about the epidemiological data.
- In addition, the authors should explain about the changes of situation form the year they obtained the data to now (with reliable reference). It can support they use the data which obtained over 10 years ago.
Methods:
- The section is too brief. The authors should explain about the study in detail.
- No statistical methods section in the Methods.
- Did the authors perform a post-hoc sample size calculation? It is informative for the potential readers.
Round 2
Reviewer 3 Report
Thank you for giving me the opportunity to review the article. The authors revised the manuscript partially. However, crucial problems still existed in the manuscript. Therefore, the reviewer thought that the manuscript cannot be published in the journal IJERPH.
Comments:
Abstract:
- The authors used the data which obtained over 10 years ago. This is a major limitation of this study. Therefore, the authors should mention about the needs of future investigations using an updated data.
AR: We have added this information in the Discussion part of the manuscript, but not in the Abstract due to the word limitation.
AC: This is important point which should be mentioned in the Abstract. The authors could mention that by rewriting the text and adjusting the word count.
Introduction:
- In addition, the authors should explain about the changes of situation from the year they obtained the data to now (with reliable reference). It can support they use the data which obtained over 10 years ago.
AR: To perform prospective cohort studies and to collect longitudinal data takes time. The Stockholm Public Health Cohort has extensive data which has been the base for many publications, and to analyze and write also takes time. In this study we aim to analyze the associations between having a non-preferred work and the incidence of spinal pain, psychological distress, and spinal pain with concurrent psychological distress, and if associations are modified by sleep disturbance. It is not to report of the occurrence of the exposures and the outcome. We have no reason to believe that the identified associations found in this study would be different today in comparison with 2010. We have added a short discussion about this in the Discussion part of the paper.
AC: The authors did not explain about the changes of situation from the year they obtained the data to now (with reliable reference). They may need more time to search the references, and to update the manuscript.
Methods:
- The section is too brief. The authors should explain about the study in detail.
- No statistical methods section in the Methods.
AR: Thank you for pointing this out. In the transfer of the manuscript into the template of the journal, we missed the section “Statistical analyses”. We have now added this to the manuscript.
AC: The authors updated the manuscript, and mentioned about the sensitivity analysis. However, there were no results in the manuscript (“showed similar results” is inappropriate). They should check the articles of observational studies, and the reporting guideline (STROBE guideline). After that, they can improve the quality of the manuscript enough to submit to other journals. - Did the authors perform a post-hoc sample size calculation? It is informative for the potential readers.
AR: No, we did not.
AC: When the authors will update the manuscript, the reviewer recommend calculating it (for potential peer-reviewers and the readers).
